biomaterials/materials science/microbiology

smart polymers, branched polymer, fungi

**Author for correspondence:**
Stephen Rimmer
e-mail: s.rimmer@bradford.ac.uk

This article has been edited by the Royal Society of Chemistry, including the commissioning, peer review process and editorial aspects up to the point of acceptance.

# Branched amphotericin functional poly(*N-iso*propyl acrylamide): an antifungal polymer

Thomas Swift[1], Emily Caseley[1], Abbigail Pinnock[2,3], Joanna Shepherd[2], Nagaveni Shivshetty[4], Prashant Garg[4], C. W. Ian Douglas[2], Sheila MacNeil[3] and Stephen Rimmer[1]

[1]Polymer and Biomaterial Chemistry Laboratories, School of Chemistry and Biosciences, University of Bradford, Bradford BD7 1DP, UK
[2]School of Dentistry, University of Sheffield, Sheffield S10 2TA, UK
[3]Department of Materials Science and Engineering, Kroto Research Institute, University of Sheffield, Sheffield S3 7HQ, UK
[4]LV Prasad Eye Institute, Banjara Hills, Hyderabad 500034, India

EC, 0000-0001-7591-143X; SR, 0000-0002-1048-1974

Branched poly(*N-iso*propylacrylamide) was functionalized with Amphotericin B (AmB) at the chain ends to produce an antifungal material. The polymer showed antifungal properties against AmB-sensitive strains of *Candida albicans*, *Fusarium keratoplasticum* and *Aspergillus flavus* (minimal inhibitory concentration ranged from 5 to 500 µg ml$^{-1}$) but was not effective against an AmB resistant strain of *C. albicans* nor against *Candida tropicalis*. The polymer end groups bound to the AmB target, ergosterol, and the fluorescence spectrum of a dye used as a solvatochromic probe, Nile red, was blue shifted indicating that segments of the polymer became desolvated on binding. The polymer was less toxic to corneal and renal epithelial cells and explanted corneal tissue than the free drug. Also, the polymer did not induce reactive oxygen species release from peripheral blood mononuclear cells, nor did it cause a substantial release of the proinflammatory cytokines, tumour necrosis factor-α and interleukin-1β (at 0.5 mg ml$^{-1}$).

## 1. Introduction

Although several antibacterial polymer systems have been developed [1–4], there has so far been less interest in the development of fungicidal polymers. However, fungal infections

are of great importance owing to complications in other diseases [5]. For example, recent studies (in Canadian intensive care units) showed up to 50% of critically ill patients were colonized with *Candida* sp. [6,7] and mortality rates as high as 60% in nocosomial infections have been reported [6,8,9]. Fungal infections are also of particular concern for immune-compromised patients [10,11] and in tropical climates, fungal infections are common, resulting in significant problems in eye infections [12,13]. However, the current pipeline for the development of antifungal agents with novel mechanisms is lacking [14].

Amphotericin B (AmB), is used as an antifungal drug but it is toxic to human tissue [15,16] particularly causing nepthrotoxicity [17,18]. The mode of action of AmB involves binding to ergosterol by providing face-to-face contacts between the macrolide and the ergosterol tetracyclic core, resulting in the creation of transmembrane channels, leading to the leakage of intracellular components and subsequent cell death [19].

Conventionally AmB is delivered as the deoxycholate complex (Fungizone) but this is highly toxic, limiting its usage to life-threatening fungal infections [20]. Lipid formulations provide lower toxicity while retaining similar efficacy to the deoxycholate complex [21]. However, significant numbers (10–25%) of patients still develop nephrotoxicity [22]. Recent attempts to formulate optimal dosage have included attaching the drug to nanoparticles [23,24], delivery within polymer micelles [25,26], conjugation to poly(ethylene glycol) chain ends [27,28], and modifying with sugar moieties [16,29–32]. In this study, we show that AmB attached to a highly branched polymer had antifungal activity but its toxicity to human cells *in vitro* was significantly reduced.

Previously, we have shown that highly branched poly(*N-iso*propylacrylamide)s (HB-PNIPAM)s with either vancomycin [33,34] or polymyxin [35] at the chain ends desolvate on binding Gram-positive or Gram-negative bacteria, respectively. In this first example of combining a branched polymer with fungi-binding end groups, we attached AmB to HB-PNIPAM. Currently, only a few polymeric conjugates with AmB are known. For example a poly(styrene-co-maleic anhydride)-AmB material was active against *Saccharomyces cerevisiae* but with three times lower efficacy than fungizone or AmB [36], while pectin conjugates showed lower activity against *Candida albicans* [32]. We show also, *in vitro*, that the polymer exhibits much-decreased toxicity to epithelial cells. To facilitate pre-clinical biological evaluation these studies, peripheral blood mononuclear cells (PBMCs), comprised of T and B cells, natural killer cells and monocytes, are widely used as clinically relevant models of the human cellular response. There are no studies reporting the response of PBMCs or purified immune cells to poly(*N-iso*propyl acrylamide) (PNIPAM) or copolymers. However, changes in cytokine production have been reported *in vivo* when copolymer composition was changed in poly(NIPAM-*co*-acrylic acid) statistical copolymer microgels [37]. Also, Fan *et al.* [38] showed that murine macrophage-like RAW264.7 produced low amounts of the cytokine interleukin-1β (IL-1β) after 48 h on PNIPAM copolymer surfaces. Proinflammatory cytokines, such as IL-1β and tumour necrosis factor-α (TNF-α), are critical mediators of the inflammatory response, and as such these cytokines are key indicators of the inflammatory response.

# 2. Results

## 2.1. Synthesis and characterization of highly branched poly(*N-iso*propylacrylamide)-Amphotericin B

The structure of AmB is shown in figure 1*a*, with linkage to the polymer via amidation of carboxylic acid functional HB-PNIPAM. HB-PNIPAM with a high concentration of carboxylic acid end groups was synthesized by the method reported previously [39], using self-condensing vinyl reversible addition-fragmentation transfer polymerization in the presence of a vinyl functional benzyl dithioate ester (4-vinylbenzyl-pyrrolecarbodithioate, VPC), which acted as a branching agent. The polymerization produced polymer with pyrrole end groups (HB-PNIPAM-Py). The ratio of NIPAM : VPC in the monomer feed was 25 : 1 and the final conversion of monomer was 96%. The chain ends were then modified to carboxylic acid and amidated with AmB via activation of the end groups as the succinimidyl ester. The reaction was carried out at pH 11 to fully solubilize the AmB.

The HB-PNIPAM-AmB was purified by repeated precipitation into diethyl ether, and then ultrafiltration in water through a 10 kDa membrane filter to retain only high molar mass material. The feed, molar mass averages and functionality are set out in the table in figure 1. The size exclusion chromatography data, shown in figure 1*b*, are derived from the refractive index (RI) and ultraviolet visible (UV/Vis) detectors. The UV/vis data were collected at 405 nm, $\lambda_{max}$ for AmB and showed that the polymer contained no AmB that was not covalently attached to the chain end. This result was confirmed using electrospray mass spectrometry with a detection limit of 0.27 µmol dm$^{-3}$ (see the electronic supplementary material, figures

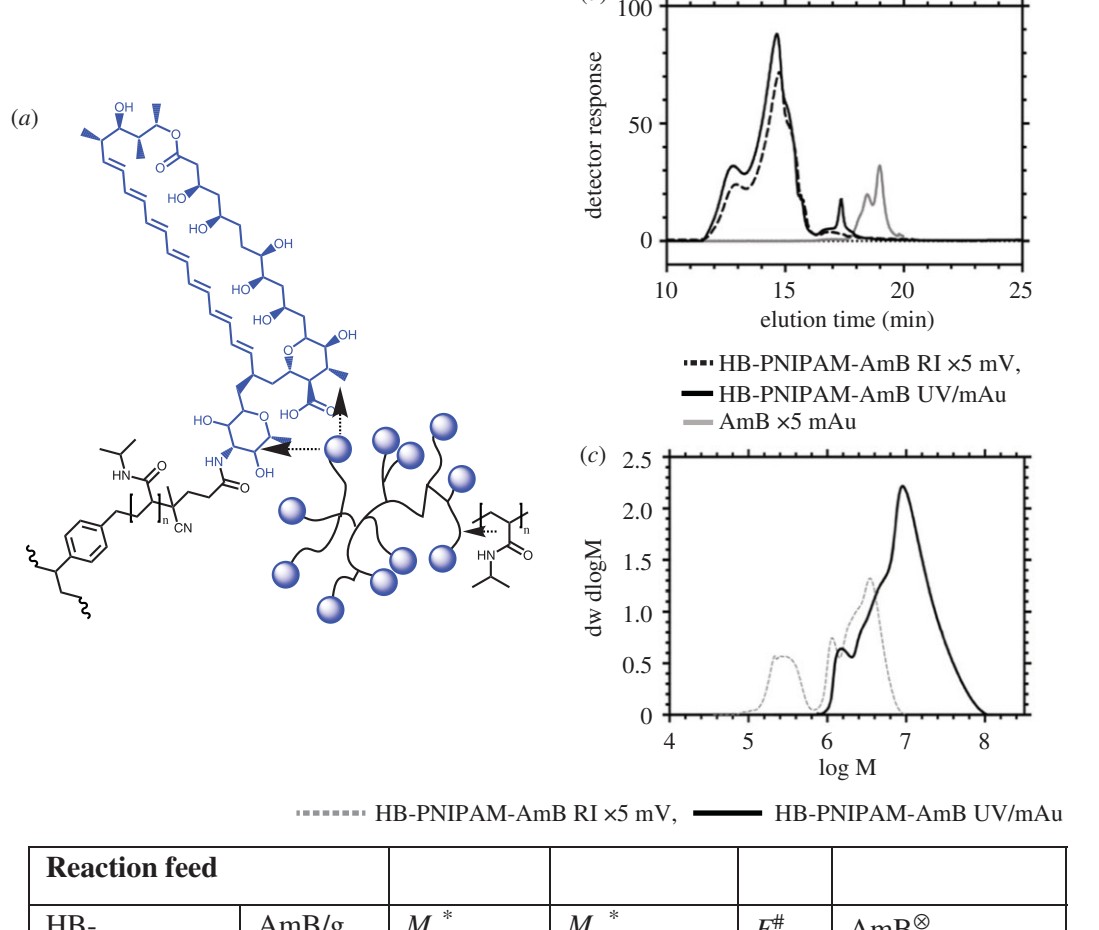

| Reaction feed | | | | | |
|---|---|---|---|---|---|
| HB-PNIPAM/g | AmB/g | $M_n^*$ (kg mol$^{-1}$) | $M_w^*$ (kg mol$^{-1}$) | $F^\#$ | AmB$^\otimes$ (mmol g$^{-1}$) |
| 1.00 | 0.03 | 8534 | 8990 | 75% | 0.25 |

**Figure 1.** Synthesis of HB-PNIPAM-AmB. (*a*) Schematic diagram showing a generalized structure of HB-PNIPAM with chain end functionality (blue) and a chain end segment with an amphotericin end group. SEC data: (*b*) RI and UV data for HB-PNIPAM-AmB and AmB. (*c*) Molar mass distributions of as prepared polymer (HB-PNIPAM-Py) and HB-PNIPAM-AmB. Asterisk denotes molar masses were obtained by size exclusion chromatography in methanol. Hash indicates functionality (*F*) expressed as molar percentage of chain ends carrying an AmB moiety determined from $^1$H nuclear magnetic resonance (NMR). Circled times symbol represents amount of AmB per unit mass of polymer derived from UV absorbance at 405 nm (extinction coefficient = $9 \times 10^{11}$ mol$^{-1}$ cm$^{-1}$).

S1–S4), which is far below the minimum inhibitory concentration (MIC = 4.68–150 µmol dm$^{-3}$ [26]) for AmB. HB-PNIPAM-AmB produced by this procedure and after purification was a high molar mass material with broad dispersity (figure 1*c*). The presence of AmB on polymer chains was demonstrated by $^1$H (NMR spectra are provided in the electronic supplementary material) and the functionalization with AmB was demonstrated by the peaks at between 4 and 6.5 ppm from the newly incorporated polyene ring (electronic supplementary material, figure S5). Diffusion-ordered spectroscopy (DOSY) NMR (electronic supplementary material, figures S6 and S7) was also used to reveal the polymer hydrodynamic radii ($R_H$) distribution in dimethyl sulphoxide (DMSO) [40]. The number and weight average hydrodynamic radii were $R_{H,n} = 2.55$ nm and $R_{H,w} = 2.60$ nm. The table shows the reactant feeds in the functionalization reaction, molar masses and functionalities.

## 2.2. Highly branched poly(*N-iso*propylacrylamide)-Amphotericin B responds to binding ergosterol

The polymers were responsive to temperature and showed a coil-to-globule ($T_{c-g}$) desolvation. $T_{c-g}$ of these polymers determined by calorimetry (electronic supplementary material, figure S8) was 36.2°C

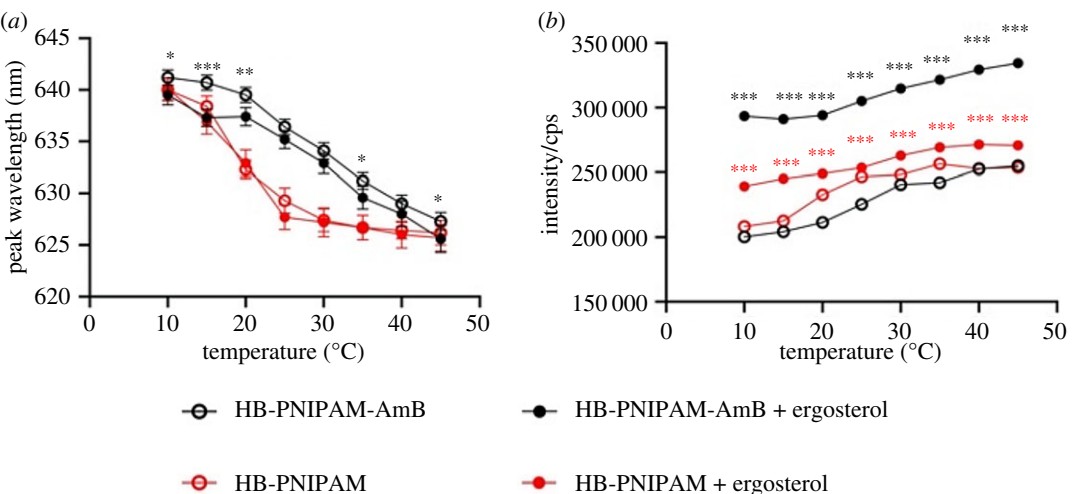

**Figure 2.** (a) Peak wavelength and (b) emission intensity of Nile red mixed with HB-PNIPAM-AmB alone (open circle) or with equivalent molar amounts of ergosterol (filled circle) or with HB-PNIPAM-Py alone (open red circle) or with equivalent molar amounts of ergosterol (filled red circle). Polymer concentrations = 1 mg ml$^{-1}$ in aqueous solution. Asterisks indicate significance: asterisks, between comparisons of HB-PNIPAM-AmB; asterisks, between comparisons of HB-PNIPAM-Py; *$p < 0.05$, **$p < 0.01$, ***$p < 0.001$.

in aqueous media. The functionalization with AmB led to an increase in $T_{c-g}$; from a peak of 18.9°C for the as prepared pyrrole-functional material. The desolvation at $T_{c-g}$ can be monitored using the addition of solvatochromic fluorescent dyes such as Nile red [39]. The emission wavelength of the Nile red dye reflects the average internal polarity of the chain segments between the swollen and desolvated forms. Decreasing the peak wavelength of the emission spectra indicated increasing hydrophobicity (desolvation) of the environment. A solution of HB-PNIPAM-AmB was mixed with Nile red and studied both in the presence and absence of ergosterol across a temperature range 5–45°C (figure 2 and electronic supplementary material, figure S9). The data (figure 2a) showed a blue shift in the peak emission wavelength of Nile red with temperature ($p < 0.001$) in both the presence and absence of ergosterol when the dye was mixed with HB-PNIPAM-AmB ($p < 0.001$). However, at all temperatures, the spectra shifted to lower wavelength on the addition of ergosterol. Figure 2b shows that the fluorescence intensity also increased as the temperature increased and that at all temperatures the peak intensity was higher in the presence of ergosterol ($p < 0.001$). Figure 2 also shows data from similar experiments carried out with the HB-PNIPAM-Py. Figure 2a shows that there was no significant shift in peak emission wavelength on adding ergosterol. Also, figure 2b shows that although there were increases in fluorescence intensity, owing to non-specific binding, the magnitude of the change was much less than when ergosterol was bound to HB-PNIPAM-AmB. Thus, the data indicated that ergosterol was bound to the AmB ligands at the chain ends of HB-PNIPAM-AmB and this binding induced a desolvation that reduced the average polarity of the environment into which Nile red was portioned.

Figure 2a shows that there is a blue shift of approximately 20 nm in the spectra as both HB-PNIPAM-AmB and HB-PNIPAM-Py pass the through $T_{c-g}$. Binding of ergosterol produces a small but consistent (across all temperatures) blue shift in HB-PNIPAM-AmB. Figure 2b shows that for HB-PNIPAM-AmB the blue shift is accompanied with a significant ($p < 0.001$) increase in peak fluorescence intensity.

## 2.3. Highly branched poly(N-isopropylacrylamide)-Amphotericin B is active against fungi

Next, AmB and HB-PNIPAM-AmB were incubated with AmB sensitive and non-sensitive species of fungi to determine the MIC (table 1). HB-PNIPAM-Py, without the AmB functionality, was ineffective against fungi (MIC > 2500 µg ml$^{-1}$). The MIC of free AmB was similar for two different AmB-sensitive strains of *C. albicans* and there was little difference between two strains in the efficacy of the polymer against them. However, although HB-PNIPAM-AmB was effective against these strains it was less so than free AmB.

Ergosterol synthesis is impaired in both *C. albicans* ATCC200955 and *Candida tropicalis* ATCC200956 and these strains were resistant to both AmB and HB-PNIPAM-AmB up to 2500 µg ml$^{-1}$. *Fusarium keratoplasticum* is renowned for forming biofilms on contact lenses and causing corneal keratitis.

**Table 1.** MIC values for AmB, HB-PNIPAM-Py and HB-PNIPAM-AmB for two AmB sensitive (SC5314, ATCC90028) and two resistant (ATCC200956, ATCC 200955) *Candida* spp., *Aspergillus flavus* ATCC16883 and *Fusarium keratoplasticum* ATCC36031.

| | weight-based MIC[a] ($\mu g\ ml^{-1}$) | | | molar MIC[b] ($\mu mol\ \mu l^{-1}$) | |
| --- | --- | --- | --- | --- | --- |
| | AmB | HB-PNIPAM | HB-PNIPAM-AmB | AmB | HB-PNIPAM-AmB |
| *C. albicans* SC5314 | 0.4 | >2500 | 4.9 | 0.43 | 1.23 |
| *C. albicans* ATCC90028 | 0.5 | >2500 | 4.0 | 0.54 | 1.00 |
| *C. albicans* ATCC200955 | >2500 | >2500 | >2500 | — | — |
| *C. tropicalis* ATCC200956 | >2500 | >2500 | >2500 | — | — |
| *F. keratoplasticum* ATCC 36031 | 0.2 | >2500 | 6.25 | 0.216 | 1.56 |
| *A. flavus* ATCC 16883 | 1.5 | >2500 | 500 | 1.62 | 125.00 |

[a]The weight based MIC is the total mass per unit volume including the polymer and the amphotercin end groups.
[b]The molar MIC is the concentration of Amphotericin ($mol\ dm^{-3}$) without regard for the attachment to the polymer.

HB-PNIPAM-AmB was also effective against this species with a similar increase in the MIC to the *C. albicans* strains that were sensitive to AmB.

Aspergillus flavus 16 883 is commonly associated with agricultural diseases but in tropical climates, it is an opportunistic pathogen that is a major cause of microbial keratitis. The data in table 1 showed that *A. flavus* 16 883 was less sensitive to AmB than the *C. albicans* strains and free AmB was more effective against this organism than HB-PNIPAM-AmB. However, the polymer with a molar MIC that was raised by approximately $\times 10^2$ compared to AmB, HB-PNIPAM-AmB did have some efficacy against this species. Therefore, these data indicate that HB-PNIPAM-AmB could be an effective therapeutic agent against some pathogenic fungi. The polymer was effective against organisms that were known to be sensitive to AmB but the MIC was substantially increased against an organism that was known to be less susceptible. Similarly, neither AmB nor HB-PNIPAM-AmB were effective against micro-organisms in which synthesis of the target (ergosterol) was impaired. It seems reasonable, therefore, to put forward the hypothesis that the action of the polymer-bound drug and the free drug operate with the same mechanism. Additionally, it is well known that AmB has low solubility in aqueous media at physiological pH. Conversely, up to $300\ mg\ ml^{-1}$ of HB-PNIPAM-AmB could be dissolved before viscous gelation started to occur.

## 2.4. Highly branched poly(N-*iso*propylacrylamide)-Amphotericin B has lower toxicity to epithelial cells and tissue than Amphotericin B

As a key potential use of HB-PNIPAM-AmB could be as a treatment of eye infections, cytotoxicity was examined using rabbit corneal epithelial cells and human explanted corneas. The toxicity of the polymer against human renal epithelial cells (HREPs) was also determined, given the well-reported nephrotoxicity of AmB. Figure 3*a,b* shows a dose-dependent decrease in cell viability as the concentration of AmB was increased. There was a significant reduction ($p < 0.01$) in the cell viability of cells after exposure to $10\ \mu g\ ml^{-1}$ AmB compared with the cell only control ($4.8 \pm 0.1\%$ and $100 \pm 5.2\%$, respectively), suggesting that a concentration of $10\ \mu g\ ml^{-1}$ would damage approximately 95% of the cells in the area in which it was applied. By contrast there was no significant loss in cell viability when exposed to the HB-PNIPAM-AmB even at $5\ mg\ ml^{-1}$, which is far in excess of the weight-based MIC data reported in table 1. Figure 3*c* shows the effects that increasing concentrations of AmB and similar concentrations of HB-PNIPAM-AmB had upon the human corneal cells located in the human corneas. The corneas subjected to HB-PNIPAM-AmB had minimal cell death, with survival rates of greater than 85% after 48 h exposure to concentrations of $15\ mg\ ml^{-1}$ whereas the cornea exposed to AmB showed only 10% ($p < 0.01$) survival when treated with a concentration of $2.5\ mg\ ml^{-1}$ or greater. AmB is also known to have an effect on the human renal system by two mechansims: decreased glomerular filtration rate [17] and damage to renal epithelial cells [41].

## 2.5. Response of peripheral blood mononuclear cells

We, therefore, also examined the effect of the polymer on primary HREPs. Figure 3*d,e* shows the cell viability of HREPs exposed to AmB and HB-PNIPAM-AmB. When exposed to $100\ \mu g\ ml^{-1}$ AmB the

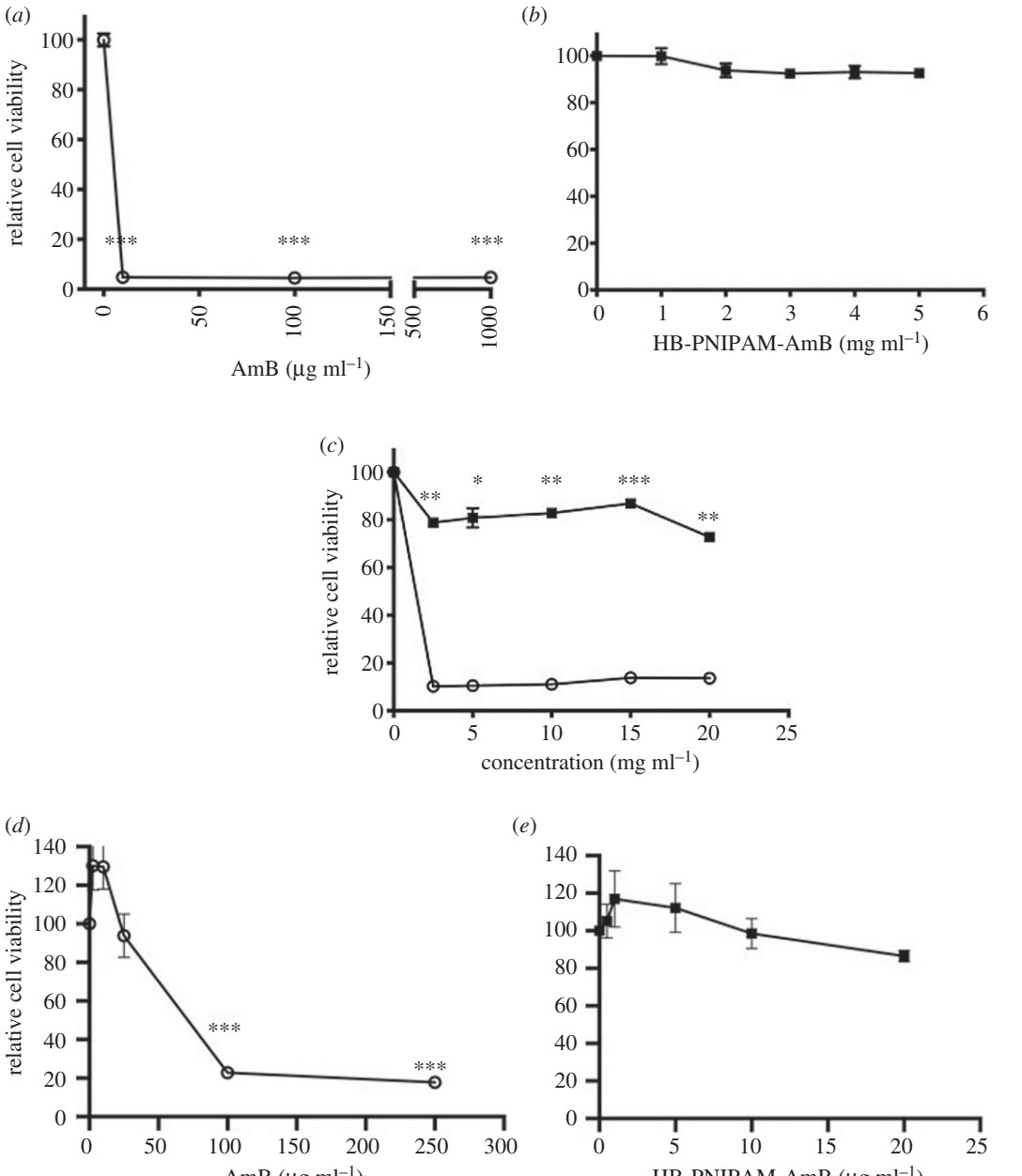

**Figure 3.** Cell viability, relative to untreated controls: (*a*) AmB and (*b*) HB-PNIPAM-AmB with rabbit limbal epithelial cells. (*c*) Human cells in explanted human cornea, (*d*) AmB and (*e*) HB-PNIPAM-AmB with human renal epithelial cells (MTT). Data obtained after exposure to HB-PNIPAM-AmB (filled square) or AmB (open circle) for 48 h. Error bars are standard error of the mean ($n = 3$). All concentrations of HB-PNIPAM-AmB are the weight of total polymer per unit volume. The data show that AmB is more toxic to epithelial cells and tissue than the polymer. Significance: (*a,b,e,d*) indicated between treated and untreated cells; (*c*) between HB-PNIPAM-AmB and AmB treated; $^*p < 0.05$, $^{**}p < 0.01$, $^{***}p < 0.001$.

cell viability fell to 20% compared to the control ($p < 0.001$). On the other hand, there was no significant difference between the cells exposed to 20 mg ml$^{-1}$ of HB-PNIPAM-AmB and untreated controls. The results suggest that when AmB is attached to the polymer there is significantly less cytotoxicity compared with AmB alone. In order to evaluate the biological impact of this polymer on the immune response *in vitro*, we assessed key indicators including metabolic activity, oxidative stress and inflammatory cytokine release from four individual human donors of PBMCs.

PBMCs do not proliferate in culture without stimulation and the MTT assay can be used to monitor metabolic activity as a result of activation [42]. Figure 4 shows how the relative metabolic activity (optical density relative to cells without treatment) changed from samples derived from the four donors. The data show that the activity of the PBMCs varied between donors but activity was higher than the positive

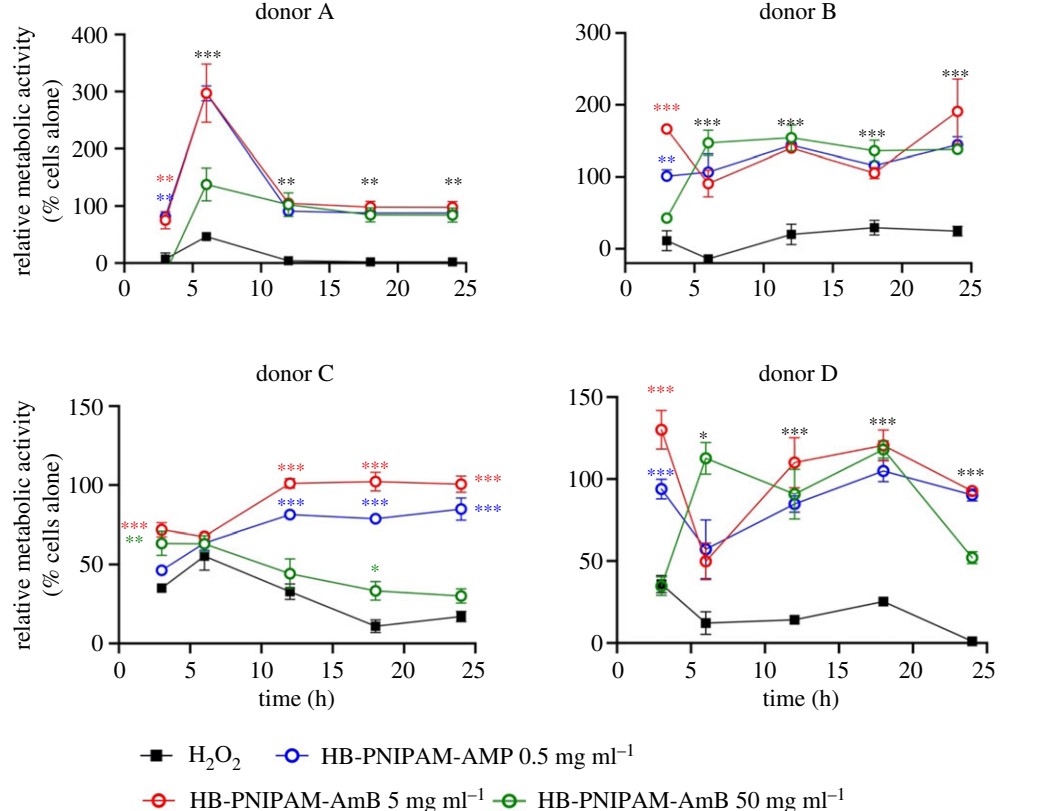

**Figure 4.** Relative metabolic activity (MTT, optical density of sample/optical density of untreated control) of PBMCs in contact with HB-PNIPAM-AmB. PBMCs derived from four anonymous donors. The data are relative to cell only controls and compared to $H_2O_2$ positive controls. Asterisks indicate significance. The black asterisks indicate that all of the HB-PNIPAM-AmB concentrations were significantly different to the $H_2O_2$ control. The brown asterisks, red asterisks and blue asterisks indicate that individual concentrations of HB-PNIPAM-AmB were significantly different to the $H_2O_2$. $^*p < 0.05$, $^{**}p < 0.01$, $^{***}p < 0.001$.

control ($H_2O_2$) for all samples. Relative metabolic activity from donors A and B remained at levels that were similar or higher than the negative control (untreated PBMCs). Increased metabolic activity is an indication of activation.

However, cells from donors C and D showed reduced activity, which probably is an indication of reduced numbers of cells, at the highest concentration (50 mg ml$^{-1}$) of HB-PNIPAM-AmB. Previously, we showed that HB-PNIPAM can be endocytosed by fibroblasts [43] and it is known that endocytosis of biomaterials can induce oxidative stress as mitochondrial disruption can lead to reactive oxygen species (ROS) release into the cytoplasm [44]. Excess ROS produced by mitochondria are important in cell signalling [45] but excess ROS result in oxidative stress. Monitoring ROS, therefore, is a useful indicator of cytotoxicity.

Figure 5 shows that there was no significant release of ROS, compared to the untreated controls, when HB-PNIPAM-AmB was in contact with PBMCs for 24 h. PBMCs release TNF-α and IL-1$\beta$, within a few hours, in response to stimulation by LPS [46,47] and AmB has also been shown to increase levels of these proinflammatory cytokines [48]. Figures 6 and 7 show data for the release of TNF-α and IL-1$\beta$ also determined over 24 h in PMBCs from four donors. The positive control was lipopolysaccharide (LPS) and untreated cells were used as negative controls.

Figure 6 shows that the PBMCs released TNF-α in varying amounts over the 24 h period. The data showed that the cells exposed to LPS released large amounts of this cytokine, which remained relatively high after 24 h. Untreated cells did not release TNF-α. Among the four donors, two different types of behaviour were observed. Firstly, there was no significant difference between cells treated with HB-PNIPAM-AmB from donors A and D and the non-treated cells. However, cells from donors B and C released significant amounts of this cytokine. In each sample, the amount released peaked over the 24 h time period and then diminished.

It was not possible in these data to discern a dose response effect and a similar observation was reported recently by Batista-Durharte et al. [49] in their studies on a AmB-B/Cochleate system. In the

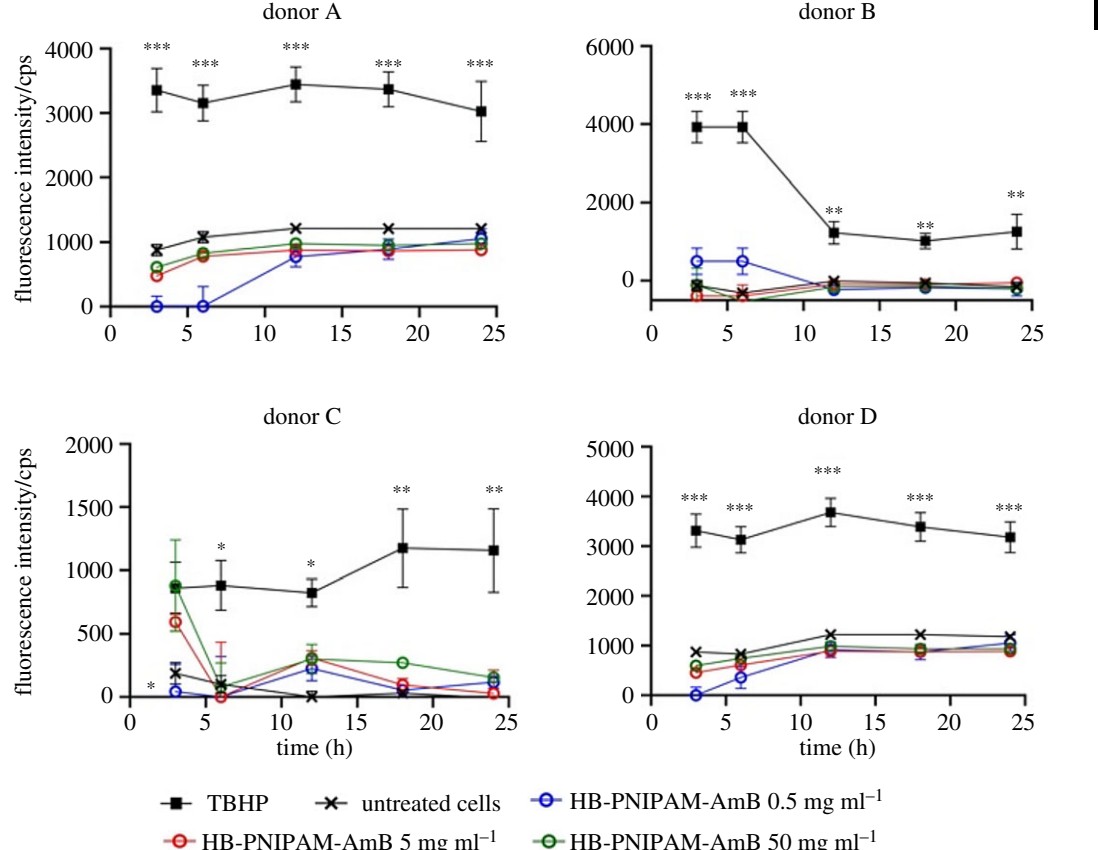

**Figure 5.** Release of reactive oxygen species in contact with HB-PNIPAM-AmB. Asterisks indicate significance relative to positive control (TBHP): $^*p < 0.05$; $^{**}p < 0.01$; $^{***}p < 0.001$.

cells from donors B and C the temporal response to HB-PNIPAM-AmB was greater than to LPS; at various time points larger amounts of TNF-α were released after treatment with HB-PNIPAM-AmB than with LPS.

However, in the cells from each donor, the HB-PNIPAM-AmB at the lowest concentration (0.5 mg ml$^{-1}$) induced release of TNF-α that was always significantly lower than the LPS controls after 12 h. Despite the response in cells from two of the donors, after 24 h the amounts of TNF-α in the supernatants was lower than the LPS controls ($p < 0.05$).

Figure 7 shows the pattern of IL-1$\beta$ release from the PBMCs. The cells treated with LPS released increasing amounts of this cytokine over time while the non-treated cells released negligible amounts. The pattern of release of IL-1$\beta$ was also variable between donors. However, a clear and consistent pattern was observed such that there were no significant increases in the release of IL-1$\beta$ when the cells were treated with 0.5 mg ml$^{-1}$ of HB-PNIPAM-AmB. On the other hand concentrations of HB-PNIPAM-AmB above 0.5 mg ml$^{-1}$ stimulated the release of IL-1$\beta$ at levels similar to or higher than that with the LPS positive control. Also, at these concentrations (above 0.5 mg ml$^{-1}$) in the cells derived from all donors, the release at the end of the experiment was higher ($p < 0.05$) or equal to cells treated with LPS. Thus, the data show minimal stimulation of the release of this cytokine up to a concentration of 0.5 mg ml$^{-1}$ but significant release above this concentration.

## 3. Discussion

Following our previous work on HB-PNIPAMs that showed beneficial responses against bacteria [33–35,39,50,51], a similar polymer with end groups that bind to fungi was synthesized. This was achieved by amidation of the polymer carboxylic acid end groups with the mycosamine group on AmB. The polymer was a high molar mass material with approximately 75% of the available end groups functionalized with AmB (the material contained 0.25 mmol g$^{-1}$ of AmB).

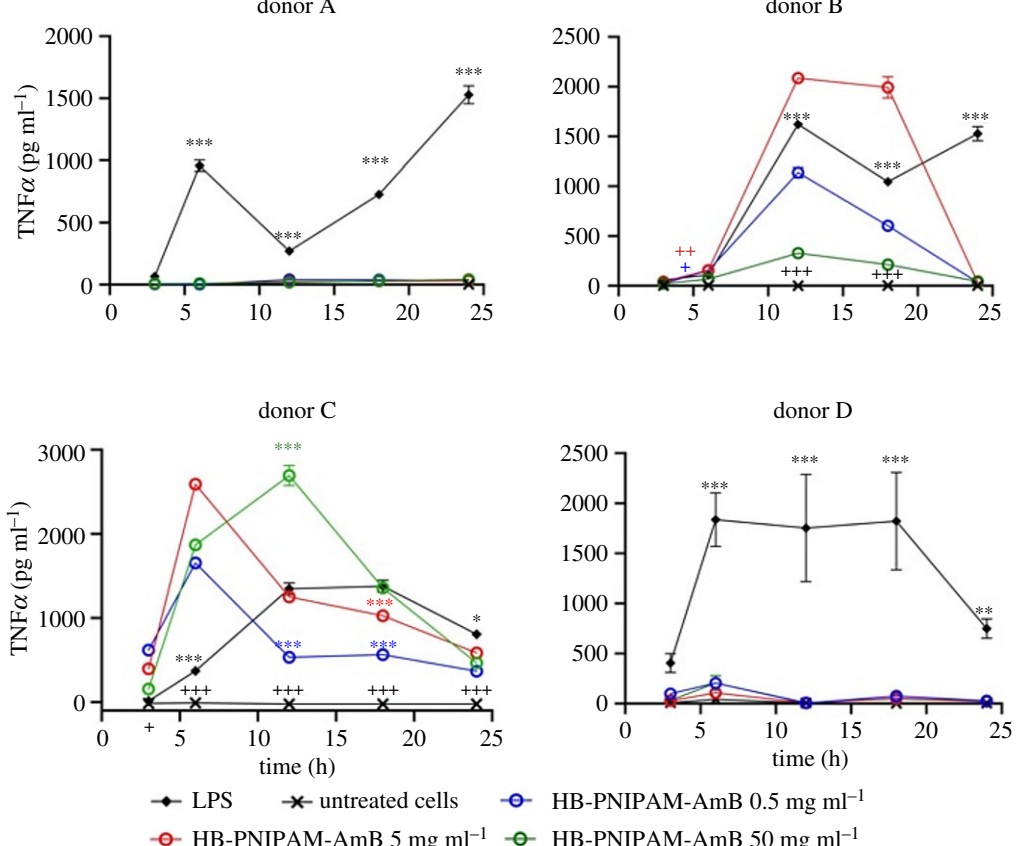

**Figure 6.** Release of TNF-α following treatment with HB-PNIPAM-AmB concentrations of cytokines in supernatants (cells seeded at 20 000 cells per well) up to 24 h after treatment with HB-PNIPAM-AmB or LPS. Asterisks and plus signs indicate significance. The black asterisks indicate that all of the HB-PNIPAM-AmB concentrations were significantly different to the LPS control at the stated time point. The brown asterisks, red asterisks and blue asterisks indicate significant differences between individual concentrations of HB-PNIPAM-AmB with LPS. $^*p < 0.05$, $^{**}p < 0.01$, $^{***}p < 0.001$. The black plus signs indicate that all of the HB-PNIPAM-AmB were significantly different to the untreated cells at the stated time points. The brown plus, red plus and blue plus signs indicate significant differences between individual concentrations of HB-PNIPAM-AmB and LPS. $^+p < 0.05$, $^{++}p < 0.01$, $^{+++}p < 0.001$.

AmB is active against many fungi and we showed that the HB-PNIPAM-AmB was also active against *C. albicans* strains that were sensitive to AmB (SC5314 and ATCC90028). Similarly, both AmB and HB-PNIPAM-AmB had similar MICs against *F. keratoplasticum* ATCC 36031. The polymer was also partially effective against *A. flavus* ATCC 16883 but with a higher MIC than against the other species. Both AmB and HB-PNIPAM-AmB were shown to be not effective against *C. albicans* ATCC200955 and *C. tropicalis* ATCC200956, which are strains known to be not sensitive to AmB. HB-PNIPAM-AmB, therefore, was shown to have similar but lower activity to AmB.

AmB acts by binding to ergosterol to form channels in the fungal cell membrane. Ergosterol also bound to the AmB end groups attached to HB-PNIPAM-AmB and polymer segments became desolvated on binding. The channels form as a result of the amphiphilic nature of AmB. In transmembrane channels, the hydrophobic polyene segments bind to ergosterol in the lipid environment of the cell membrane and the polyol segments face an aqueous phase forming a transmembrane pore, composed of single or double lengths of the AmB : ergosterol complex [52]. However, at lower concentrations 'non-aqueous' pores are formed, which do not cross the membrane but in which AmB is inserted into the bi-layer [53]. Ergosterol sits within the cell membrane beneath the cell wall. The cell wall of fungi is porous and in studies on yeast (*S. cerevisiae*) with linear polymers of various sizes, it has been shown that macromolecules with $R_H$ of up to 5.8 nm can permeate through [54] and much larger deformable vesicles have also been shown to deliver AmB to the inner membrane [55]. The cell membrane is thus accessible to polymeric drugs. The data presented here showed that HB-PNIPAM-AmB both bound to ergosterol and was effective against a range of bacteria. Given that HB-PNIPAM-AmB had a $R_H$ of 2.6 nm it seems reasonable that this high

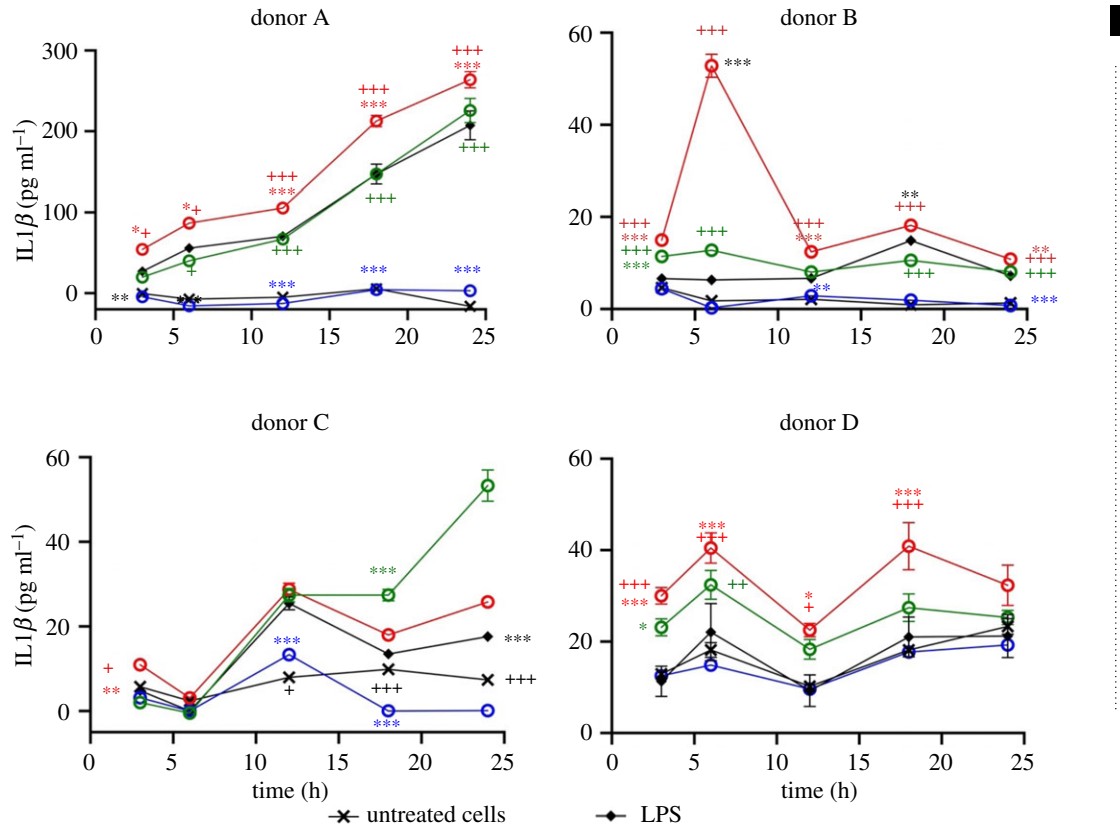

**Figure 7.** Release of IL-1β following treatment with HB-PNIPAM-AmB concentrations of IL-1β in supernatants (cells seeded at 20 000 cells per well) up to 24 h after treatment with HB-PNIPAM-AmB or LPS. There is no significant increase in the release of IL-1β following treatment with 0.5 mg ml$^{-1}$ of HB-PNIPAM-AmB but concentrations of over 5 mg ml$^{-1}$ stimulated release of increased amounts of this cytokine. Asterisks and plus signs indicate significance. The black asterisks indicate that all of the HB-PNIPAM-AmB concentrations were significantly different to the LPS control at the stated time point. The brown asterisks, red asterisks and blue asterisks indicate significant differences between individual concentrations of HB-PNIPAM-AmB with LPS. $^*p < 0.05$, $^{**}p < 0.01$, $^{***}p < 0.001$. The black plus signs indicate that all of the HB-PNIPAM-AmB were significantly different to the untreated cells at the stated time points. The brown plus, red plus and blue plus signs indicate significant differences between individual concentrations of HB-PNIPAM-AmB and LPS. $^+p < 0.05$, $^{++}p < 0.01$, $^{+++}p < 0.001$.

molar mass but compact branched polymer would also diffuse through the cell wall to reach the ergosterol targets.

The polymer reported in this current study here had similar but lower activity to free AmB but much lower toxicity to two types of epithelial cells and cells in the explanted cornea. AmB was attached to the HB-PNIPAM via the amine group, which is adjacent to the C2′ hydroxyl group. The C2′ hydroxyl group is also involved in binding to cholesterol. Binding to cholesterol in human cells has been implicated in the toxicity of AmB and Brandon *et al.* [56] showed that deleting the C2′ hydroxyl group minimized this interaction while binding to ergosterol was maintained. More work is required to fully determine the mechanism but it seems reasonable that the close proximity of the polymer segments to C2′ could have similar effects to deletion, possibly owing to competitive hydrogen bonding between the C2′ hydroxyl and the acrylamide units. One application under consideration for these materials is to treat fungal keratitis. The ocular environment is immune privileged and the minimal toxicity to cornel epithelial cells in explanted cornea provides significant progress to an effective therapeutic system for fungal infections of the eye. However, in blood-contacting applications, for example in wounds infected with fungi, it is important to consider the response of cells of the immune system. The *in vivo* response to a material involves interaction with proteins and immune cells [57] and here we chose to study the response of PBMCs over 24 h. The cells of the immune system undergo oxidative bursts of ROS, which is part of the response to microbes. It was shown that HB-PNIPAM-AmB over 24 h did not generate ROS in the PBMCs when compared to untreated cells and the data indicate minimal ROS effects in PBMCs in response to HB-PNIPAM-AmB. The response of immune cells is mediated by a

range of cytokines. In this respect, TNF-$\alpha$ and IL-1$\beta$ are important in the inflammatory response and they are implicated in the adverse effects of AmB [58]. HB-PNIPAM-AmB did initiate a release of these inflammatory cytokines but TNF$\alpha$ was reduced to levels similar to the untreated controls after 24 h and the cells did not release IL-1$\beta$ at a concentration of 0.5 mg ml$^{-1}$. Thus, the data justify further considerations aimed at using HB-PNIPAM-AmB as a low toxicity variant of AmB.

# 4. Experimental

## 4.1. Synthesis

### 4.1.1. Synthesis of highly branched poly(N-isopropylacrylamide)-Amphotericin B

HB-PNIPAM with COOH end groups (HB-PNIPAM-COOH) was prepared as previously described [39]. The detail of the synthesis is described in §2 of the supporting information. The HB-PNIPAM-COOH (5.00 g) was dissolved in dimethyl formamide (DMF) (55 ml). To this, a solution of N-Hydroxysuccinimide (0.858 g, 7.46 mmol) and N-N-Dicyclohexylcarbodimide (DCC) (1.539 g, 7.46 mmol) in DMF (15 ml) was added. The mixture was stirred under N$_2$ overnight and all solid products removed via gravity filtration. The remaining solution was precipitated into diethyl ether, dissolved in ethanol and concentrated via ultrafiltration three times. The remaining solvent was removed by rotary-evaporation and the solid dried under vacuum at room temperature. This HB-PNIPAM with succinimide end groups (1.00 g) was dissolved in ice-cold water (50 ml). A solution of AmB (30 mg, 0.032 moles) in 0.1 M sodium phosphate buffer (pH 8.5, 10 ml) and water (10 ml) was added to the polymer solution (1.0 g). The solution pH was increased to pH 11 (by addition of 0.01 M NaOH) while stirring on ice overnight, then at room temperature for 24 h. The solution was ultrafiltered in water using 10 kDa pore membrane filters in water at pH 11 (250 ml extracted—repeated seven times), then once again at pH 7 (250 ml extraction) before the sample was freeze-dried. The final product was a pale yellow solid (0.99 g yield) and was characterized by $^1$H NMR [400 MHz, DMSO (ppm): 1.05 (6H, s, -N(CH$_3$)$_2$) 1.55 (2H, br m, CH$_2$-CH-Ar-) 2.01 (2H, br m, -CH$_2$-CH-CO-NH-), 3.82 (H$_2$O-polymer), 7.26 (br m, -Ar-)], sulfur content (0.24%), 4.5–6.5 (multiple peaks (see the electronic supplementary material, figure S1, (C=C–C)) and DOSY ($R_{Hn}$ = 2.55 nm, $R_{Hw}$ = 2.66 nm) and stored at −18°C.

## 4.2. Solubility of highly branched poly(N-isopropylacrylamide)-Amphotericin B

Solutions of up to 300 mg ml$^{-1}$ of HB-PNIPAM-AmB were found to be stable for 24 h on a workbench. In the same timeframe solutions of 400 mg ml$^{-1}$ fully dissolved but became a viscous gel.

## 4.3. Fluorescence dye studies of polymer solutions

A stock solution of Nile red was prepared (0.4 mg cm$^{-3}$, DMSO). This was diluted (to 10$^{-7}$ mol dm$^{-3}$) with ultrapure water. HB-PNIPAM polymers (11 mg) were dissolved in ultrapure water (7 ml) and Nile red stock solution (100 µl) was added. The fluorescence spectrum was then recorded on a Horiba Fluoromax-4 excitation 580 nm, emission 560–800 nm, with slit widths of 1 nm. Peak wavelength emission and intensity was calculated from the Gaussian distribution of wavelengths. The data for peak wavelength and emission intensity were compared using 2-way ANOVA using Fishers LSD post hoc analysis.

## 4.4. Fungi

*Culture conditions for fungi. Candida albicans* cells, laboratory strain SC5314 or ATCC90028, were cultured on solid brain heart infusion (BHI) (Oxoid) medium at 37°C for 24 h and stored at 4°C for up to one month. Prior to experiments, a colony of *C. albicans* was sub-cultured into liquid BHI medium overnight at 37°C. *Candida albicans* (Robin) ATCC200955, *C. tropicalis* ATCC200956, *A. flavus* ATCC16883 and *F. keratoplasticum* ATCC36031 were similarly cultured on solid Sabrauds (Oxoid) medium at 37°C for 24 h and stored at 4°C for up to one month. Prior to experiments, colonies were sub-cultured into liquid Sabrauds medium overnight at 37°C. All liquid cultures were static apart from *A. flavus* which was incubated shaking at 100 rpm.

  *MICs for amphotericin polymers.* Overnight cultures were adjusted to an optical density at 600 nm of 0.1 and incubated with Amphotericin polymer, which was serially diluted 1 : 2 from 2500–2.44 µg ml$^{-1}$, for

16 h at 37°C in a 96 well plate. The concentration of polymer at which there was no visible growth was determined to be the MIC.

## 4.5. Cells and tissue

### 4.5.1. Isolation and culture of rabbit limbal epithelial cells

Limbal rims from corneal-scleral buttons were excised from wild brown rabbit heads, as previously described [59]. Tissue was immersed in 2.5 mg ml$^{-1}$ (w/v) dispase II solution in Dulbecco's minimum essential medium (DMEM) for 1 h at 37°C. The limbal rims were subsequently scraped gently using forceps to remove the epithelial cells. The dispase-cell suspension was centrifuged at 200 g for 5 min and resuspended in culture medium containing DMEM: Ham's F12 (1 : 1) supplemented with 10% fetal calf serum, 100 U ml$^{-1}$ penicillin and 100 U ml$^{-1}$ streptomycin, 2.5 µg ml$^{-1}$ AmB, 5 µg ml$^{-1}$ insulin and 10 ng ml$^{-1}$ epidermal growth factor. Cells were cultured with irradiated 3T3-mouse fibroblasts at a cell density of $2.4 \times 10^4$ cells cm$^{-2}$ in culture medium. Cells were not used beyond passage 3. For experiments, cells were seeded into 24 well plates at $5 \times 10^4$ cells per well and cultured for 24 h.

### 4.5.2. Rabbit limbal epithelial cell viability

Prior to determining the viability of rabbit limbal epithelial cells, cells were washed 3× with phosphate-buffered saline (PBS; 0.01 M, pH 7.4) to remove residual antimicrobials. HB-PNIPAM-AmB (20, 15, 10, 5, 4, 3, 2, 2.5, 1 and 0 mg ml$^{-1}$) or AmB (0, 1, 10, 100 and 1000 µg ml$^{-1}$ or 0, 2.5, 5, 10, 15 and 20 mg ml$^{-1}$) were dissolved in culture medium without penicillin, streptomycin or AmB, and added to pre-seeded epithelial cells for a further 24 h. Epithelial cell viability was determined using Alamar Blue reagent (5 µg ml$^{-1}$) dissolved in PBS. Cells were incubated with Alamar Blue for 30 min and the fluorescence of the solution determined at 570 nm excitation and 585 nm emission (Infinite 200, Tecan). Data for each polymer is presented as percentage viability relative to an untreated control. Each assay was performed in triplicate and data represents the mean ± s.e.m. from three independent experiments. The data were compared to the untreated cells using one-way ANOVA with Tukey *post hoc*. The variances were calculated via the propagation of errors following the calculation of the percentage viability.

### 4.5.3. Human renal epithelial cell culture

Primary HREPs purchased from Promocell (Germany) were maintained in low-serum (5% V/V) renal cell epithelial cell growth medium 2 supplemented with the associated SupplementPack. All cells were maintained at 37°C in a 5% CO$_2$ atmosphere with 95% humidity. The cell growth medium was replaced every 2–3 days and cells were passaged once confluent.

### 4.5.4. Human renal epithelial cell viability

A MTT assay (Merck) was used to assess cell viability. HREPs were plated in flat bottomed poly-d-lysine-coated 96-well plates at a density of 20 000 cells per well and incubated with AmB or HB-PNIPAM-AmB at the indicated concentrations for 24 h. Ten microlitres of MTT reagent was subsequently added to each well and the plate incubated at 37°C for 4 h. One-hundred microlitres of solubilization solution was then added per well and the cells incubated at 37°C overnight before reading the absorbance at 570 nm using a Tecan Infinite F50 plate reader running Magellan for F50 software. The data are presented as the percentage of the optical density of the cell only control. The data were compared to the untreated cells using one-way ANOVA with Tukey *post hoc*. The variances were calculated via the propagation of errors following the calculation of the percentage viability.

### 4.5.5. Human excised corneas

Cell viability tests were performed on the cadaveric human corneas which were unsuitable for transplant. These were acquired from the Ramayamma International Eye Bank, LV Prasad Eye Institute, Hyderabad, India. All corneas were obtained following procedures approved by the institutional review board for the protection of human subjects. Isolation and culture of human corneas were performed as previously described. [35]. Prior to use, corneas were washed three times with PBS and incubated in antibiotic- and antifungal-free medium for at least 24 h to remove residual antimicrobials. Polymer and AmB in a

range of concentrations (2.5, 5, 10, 15, 20 mg ml$^{-1}$) were dissolved in the culture media and added to the *ex vivo* cornea for 24 h at 37°C. The metabolic activity of the cells was measured by an Alamar Blue assay. Briefly 100 µl Alamar Blue was added to each cornea and incubated at 37°C for 2–4 h. The fluorescence was then read on plate reader (Spectra Max M3 using SoftMax Pro Data Acquisition and Analysis Software) and expressed as percentage viability compared to control cells. The data were analysed using two-way ANOVA with repeated measures and *post hoc* analysis with Sidak's procedure for comparison of two means. The data are presented as the percentage of the untreated controls. The variances were calculated via the propagation of errors following the calculation of the percentage viability.

### 4.5.6. Isolation and culture of primary blood mononuclear cells

Whole peripheral blood was taken from healthy human donors (approval 17/YH/0086) and the PBMCs isolated using Sepmate50 tubes with Lymphoprep as the density gradient medium. PBMCs were separated by centrifugation at (1200$g$) for 10 min, followed by the removal of the PBMC-containing supernatant. Cells were washed with calcium-free PBS containing 2% fetal bovine serum (FBS) three times by centrifugation at (800$g$) for 8 min. Cells were incubated for 24 hr in 5% (v/v) $CO_2$ at 37°C in RPMI 1640 culture medium (RCM) (RPMI 1640 (ThermoFisher Scientific) supplemented with GlutaMAX supplement, supplemented with 10% FBS (ThermoFisher Scientific), 2 mM L-Glutamine (ThermoFisher Scientific) and 100 U ml$^{-1}$ penicillin-streptomycin (ThermoFisher Scientific)) before the addition of polymer solutions or controls as indicated. PBMCs were incubated with the polymers for 3, 6, 12, 18 or 24 h as indicated.

### 4.5.7. Cell viability testing of primary blood mononuclear cells

Cell viability was evaluated using an MTT assay (Merck). PBMCs were plated at a density of 20 000 cells per well in flat bottomed 96-well plates and incubated with polymers at three different concentrations for 3, 6, 12, 18 or 24 h as outlined above. Following this, 10 µl of MTT reagent was added to each well and incubated at 37°C for 4 h. One-hundred microlitres of solubilization solution was then added to each well and the cells incubated at 37°C overnight before recording the absorbance at 570 nm with a Tecan Infinite F50 plate reader running Magellan for F50 software. The data were analysed using two-way ANOVA with Tukey *post hoc*. The data are presented as the percentage of the untreated controls. The variances were calculated via the propagation of errors following the calculation of the percentage viability.

### 4.5.8. Oxidative stress testing

Oxidative stress in PBMCs was determined by measuring ROS release using a DCFDA cellular ROS detection assay kit (Abcam) according to the manufacturer's instructions. Briefly, PBMCs were seeded onto black, clear-bottomed 96-well plates and incubated with polymers for the specified period of time. Eleven microlitres of the tertiary butyl peroxide positive control in phenol red-free RCM was added to the relevant wells for a final concentration of 50 µM and the cells incubated at 37°C for 3 h 45 min. Cells were then treated with 25 µM DCFDA in phenol red-free RCM and incubated at 37°C for 45 min and the fluorescence immediately measured using a Promega Explorer microplate reader with the excitation and emission wavelengths of 475 nm and 500–550 nm, respectively. The data were analysed using two-way ANOVA with Dunnett's *post hoc* procedure for comparison to the non-treated cells control.

### 4.5.9. Measurement of inflammatory cytokine release

Measurement of TNF-α and IL-1$\beta$ in cell culture media was analysed by sandwich enzyme-linked immunosorbent assay (ELISA) (Diaclone). Assays were carried out according to the manufacturer's instructions. The data were analysed using two-way ANOVA with Tukey *post hoc* procedure.

## 5. Conclusion

In conclusion, HB-PNIPAM with AmB end groups binds to ergosterol and provides a desolvation phase transition of a fraction of polymer segments. The polymers are much less toxic than the free drug to epithelial cells. HB-PNIPAM-AmB also had relatively small effects *in vitro* on PBMCs at concentrations

that were below the MICs for *C. albicans* and *F. keratoplasticum*. The polymers are active against fungi and demonstrate therapeutically useful MICs as well as providing an increase in fluorescence intensity on binding to the membrane target, ergosterol.

Ethics. Ethical approval was obtained from University of Bradford ethics committee and the L. V. Prasad ethics committee.

Data accessibility. Provided as the electronic supplementary material.

Authors' contributions. S.R.: supervision of T.S. and E.C., wrote the paper and analysed the data. T.S.: synthesis of polymers, characterization and photo physics. E.C.: monocyte work and HREP culture. A.P.: microbiology work. N.S.: microbiology and cornea work. P.G.: supervision of N.S., clinical input, micro and cornea work. C.W.I.D.: microbiology supervision. S.M.: supervision of cell culture.

Competing interests. There are no conflicting financial interests of the authors.

Funding. The work was funded by the Welcome Trust DBT Alliance (0998800/B/12/Z) and MRC (UK) (grant no. 16038).

Acknowledgements. The authors thank Mr David Pownall for technical support of the synthesis work.

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
