## [Reviewer comments · Royal Society Open Science]

Review History

RSOS-201655.R0 (Original submission)

Review form: Reviewer 1

Is the manuscript scientifically sound in its present form?

Yes

Are the interpretations and conclusions justified by the results?

Yes

Is the language acceptable?

Yes

Do you have any ethical concerns with this paper?

No

Have you any concerns about statistical analyses in this paper?

Yes

Recommendation?

Accept with minor revision (please list in comments)

Comments to the Author(s)

This paper by Prof Rimmer and co-workers describes the utilisation of their branched PNIPAM materials as antifungals. The work is novel, the results are scientifically interesting and the findings medically important. I believe this paper will be of interest to the readers of Open Science. I do have some suggestions for revision prior to publication.

Page 3. Line 9-11- "mortality rates as high as 60% with *Candida* sp. infections." This could be rephrased to make clear that this is (presumably) a mortality rate for certain infections, rather than an estimate for the average infection of any candida type.

Page 3. Line 34 onward: The authors move to a personal tense (e.g. "We") – the editor can recommend if this should be moved to an impersonal tense.

Page 5, Line 49: The authors state that the coil-globule transition occurs at the LCST. At this point the use of LCST and Tc-g seems to vary in the manuscript. E.g. "...increase in Tc-g; from a peak of 18.9 °C for the as prepared pyrrole-functional material" compared to "LCST measurements were carried out on a Nano DSC by TA Instruments" (SI, 3.). LCST used in the methods section in the main paper. If these are used interchangeably, I think a single designation would be clearer.

Page 7, Section "HB-PNIPAM-AmB is active against fungi" – it would be useful to clarify if the MIC is calculated based on the moles of AmB in the polymer, or the moles of polymer itself. I assumed moles of AmB in the polymer-AmB solution, but then the cytotoxicity experiments, which I assumed were conducted based on polymer concentration, asked for comparison to the MICs ("even at 5 mg mL⁻¹, which is far in excess of the MIC data"). I think this confusion arises because the weight-based MIC appears to be related to polymer concentration, whereas the molar MIC is based on conc of AmB in the polymer. I guess this is the case because the only way I could get the numbers to make sense is convert weight to moles using this conversion factor given in the table in figure 1. I recommend making this conversion clear in the table headings and caption. Page 7. "AmB compared with the cell only control (... 100 ± 5.2 %)" It is not very clear to me from the methods how you determined error on these measurements of control if the data are normalised to these healthy cells.

Page 8, figure 3. The caption is written in a confusing manner. "Cell viability, relative to untreated controls, either in the absence of polymer or in the presence of AmB:" I guess these comments relate to the controls, but it is written in a confusing manner. Then for A,B and D,E the two test samples are separated, but placed together in C. I would suggest revising the figure caption and splitting element C to make this easier to follow for the reader.

Page 17. "HB-PNIPAM-AmB both bound to ergosterol and was effective against a range of bacteria" – I assume this is meant to read "fungi"

Page 18. "Synthesis of HB-PNIPAM-AmB": "HB-PNIPAM with COOH end groups prepared as previously described.³⁶" – I understand that this synthesis is described in the SI under "Synthesis of HB-PNIPAM Precursor." It would be easier to follow with a reference to the SI and a consistent terminology used for this polymer. This method could also include the LCST of the final product, for consistency.

Page 19 "Overnight cultures were adjusted to an optical density at 600 nm of 0.1 and incubated with Amphotericin polymer, which was serially diluted 1:2 from 2500-2.44 µg mL⁻¹, for 16 hours in a 96 well plat" – it would be useful to state the incubation temperature (I guess 37C), especially given that the LCST seems to be very close to 37C. I would imagine the LCST transition may affect MIC.

Page 20. "Rabbit limbal epithelial cell viability" and "Human Renal epithelial cell viability" – whether data are compared using one or two way ANOVA should be stated.

Page 21, Conclusions. "In conclusion, highly-branched poly(N-isopropyl acrylamide) with AmB end groups binds to ergosterol with a desolvation phase transition". This implies to me that the

ergosterol binding to PNIPAM-AmB induces a phase change of the whole material (i.e. the coil-globule transition), when I understand there is really some dehydration of AmB end groups. I suggest this statement is revised.

Review form: Reviewer 2 (Nick Turner)

Is the manuscript scientifically sound in its present form?

Yes

Are the interpretations and conclusions justified by the results?

Yes

Is the language acceptable?

Yes

Do you have any ethical concerns with this paper?

No

Have you any concerns about statistical analyses in this paper?

No

Recommendation?

Accept with minor revision (please list in comments)

Comments to the Author(s)

This is a really interesting paper and i enjoyed reading it. I am recommending publication with minor cosmetic revisions.

P2, L3 "less toxic" than what"? Recommend this single sentence be more specific.

Figure 1: The use of a,b,c in superscript was difficult to see and could be confusing to non-specialists (who may not understand M_W^a is actually M_W with an annotation. Recommend using *,\$,# instead.

Figure 1: Move in figure text into Figure 1 legend.

Table 1: Legend above table not below.

Figure 3: legend. $P < 0.001$ is missing its ***

Figure 4: Check colours of * vs those on figures. Consider rewording the significance sentence as it is a little confusing (suggest removing colours and just use *, **, *** in black and description) . See Figure 5 for clear example. I think reader will realise that the * are colour coded to lines in the figures. Same for other figures (6+7)>

Decision letter (RSOS-201655.R0)

Dear Miss Rimmer:

Title: Branched Amphotericin Functional Poly(N-isopropyl acrylamide): an Antifungal Polymer
Manuscript ID: RSOS-201655

Thank you for submitting the above manuscript to Royal Society Open Science. On behalf of the Editors and the Royal Society of Chemistry, I am pleased to inform you that your manuscript will be accepted for publication in Royal Society Open Science subject to minor revision in accordance with the referee suggestions. Please find the reviewers' comments at the end of this email.

The reviewers and handling editors have recommended publication, but also suggest some minor revisions to your manuscript. Therefore, I invite you to respond to the comments and revise your manuscript.

Because the schedule for publication is very tight, it is a condition of publication that you submit the revised version of your manuscript before 15-Nov-2020. Please note that the revision deadline will expire at 00.00am on this date. If you do not think you will be able to meet this date please let me know immediately.

Kind regards,
Dr Laura Smith
Publishing Editor, Journals

On behalf of the Subject Editor Professor Anthony Stace and the Associate Editor Professor Chaohua Cui.

RSC Associate Editor:
Comments to the Author:
(There are no comments.)

RSC Subject Editor:
Comments to the Author:
(There are no comments.)

Reviewer comments to Author:
Reviewer: 1

Comments to the Author(s)

This paper by Prof Rimmer and co-workers describes the utilisation of their branched PNIPAM materials as antifungals. The work is novel, the results are scientifically interesting and the findings medically important. I believe this paper will be of interest to the readers of Open Science. I do have some suggestions for revision prior to publication.

Page 3. Line 9-11- "mortality rates as high as 60% with *Candida* sp. infections." This could be rephrased to make clear that this is (presumably) a mortality rate for certain infections, rather than an estimate for the average infection of any candida type.

Page 3. Line 34 onward: The authors move to a personal tense (e.g. "We") - the editor can recommend if this should be moved to an impersonal tense.

Page 5, Line 49: The authors state that the coil-globule transition occurs at the LCST. At this point the use of LCST and Tc-g seems to vary in the manuscript. E.g. "...increase in Tc-g; from a peak of 18.9 °C for the as prepared pyrrole-functional material" compared to "LCST measurements were carried out on a Nano DSC by TA Instruments" (SI, 3.). LCST used in the methods section in the main paper. If these are used interchangeably, I think a single designation would be clearer.

Page 7, Section "HB-PNIPAM-AmB is active against fungi" - it would be useful to clarify if the MIC is calculated based on the moles of AmB in the polymer, or the moles of polymer itself. I assumed moles of AmB in the polymer-AmB solution, but then the cytotoxicity experiments,

which I assumed were conducted based on polymer concentration, asked for comparison to the MICs ("even at 5 mg mL⁻¹, which is far in excess of the MIC data"). I think this confusion arises because the weight-based MIC appears to be related to polymer concentration, whereas the molar MIC is based on conc of AmB in the polymer. I guess this is the case because the only way I could get the numbers to make sense is convert weight to moles using this conversion factor given in the table in figure 1. I recommend making this conversion clear in the table headings and caption. Page 7. "AmB compared with the cell only control (... 100 ± 5.2 %)" It is not very clear to me from the methods how you determined error on these measurements of control if the data are normalised to these healthy cells.

Page 8, figure 3. The caption is written in a confusing manner. "Cell viability, relative to untreated controls, either in the absence of polymer or in the presence of AmB:" I guess these comments relate to the controls, but it is written in a confusing manner. Then for A,B and D,E the two test samples are separated, but placed together in C. I would suggest revising the figure caption and splitting element C to make this easier to follow for the reader.

Page 17. "HB-PNIPAM-AmB both bound to ergosterol and was effective against a range of bacteria" - I assume this is meant to read "fungi"

Page 18. "Synthesis of HB-PNIPAM-AmB": "HB-PNIPAM with COOH end groups prepared as previously described.³⁶" - I understand that this synthesis is described in the SI under "Synthesis of HB-PNIPAM Precursor." It would be easier to follow with a reference to the SI and a consistent terminology used for this polymer. This method could also include the LCST of the final product, for consistency.

Page 19 "Overnight cultures were adjusted to an optical density at 600 nm of 0.1 and incubated with Amphotericin polymer, which was serially diluted 1:2 from 2500-2.44 µg mL⁻¹, for 16 hours in a 96 well plat" - it would be useful to state the incubation temperature (I guess 37C), especially given that the LCST seems to be very close to 37C. I would imagine the LCST transition may affect MIC.

Page 20. "Rabbit limbal epithelial cell viability" and "Human Renal epithelial cell viability" - whether data are compared using one or two way ANOVA should be stated.

Page 21, Conclusions. "In conclusion, highly-branched poly(N-isopropyl acrylamide) with AmB end groups binds to ergosterol with a desolvation phase transition". This implies to me that the ergosterol binding to PNIPAM-AmB induces a phase change of the whole material (i.e. the coil-globule transition), when I understand there is really some dehydration of AmB end groups. I suggest this statement is revised.

Reviewer: 2

Comments to the Author(s)

This is a really interesting paper and i enjoyed reading it. I am recommending publication with minor cosmetic revisions.

P2, L3 "less toxic" than what"? Recommend this single sentence be more specific.

Figure 1: The use of a,b,c in superscript was difficult to see and could be confusing to non-specialists (who may not understand M_w^a is actually M_w with an annotation. Recommend using *,\$,# instead.

Figure 1: Move in figure text into Figure 1 legend.

Table 1: Legend above table not below.

Figure 3: legend. P<0.001 is missing its ***

Figure 4: Check colours of * vs those on figures. Consider rewording the significance sentence as it is a little confusing (suggest removing colours and just use *, **, *** in black and description). See Figure 5 for clear example. I think reader will realise that the * are colour coded to lines in the figures. Same for other figures (6+7)>

Author's Response to Decision Letter for (RSOS-201655.R0)

See Appendix A.

Decision letter (RSOS-201655.R1)

Dear Professor Rimmer:

Title: Branched Amphotericin Functional Poly(N-isopropyl acrylamide): an Antifungal Polymer
Manuscript ID: RSOS-201655.R1

It is a pleasure to accept your manuscript in its current form for publication in Royal Society Open Science. The chemistry content of Royal Society Open Science is published in collaboration with the Royal Society of Chemistry.

On behalf of the Subject Editor Professor Anthony Stace and the Associate Editor Professor Chaohua Cui.

RSC Associate Editor
Comments to the Author:
(There are no comments.)

Reviewer(s)' Comments to Author:

Dear sir,

Please find attached a revision of our paper for your consideration. Our detailed response to each of the referees are set out below.

Your faithfully

Response to referees

Reviewer: 1

Comments to the Author(s)

This paper by Prof Rimmer and co-workers describes the utilisation of their branched PNIPAM materials as antifungals. The work is novel, the results are scientifically interesting and the findings medically important. I believe this paper will be of interest to the readers of Open Science. I do have some suggestions for revision prior to publication.

1. Page 3. Line 9-11- "mortality rates as high as 60% with *Candida* sp. infections." This could be rephrased to make clear that this is (presumably) a mortality rate for certain infections, rather than an estimate for the average infection of any candida type.

Answer: In one of the studies we cited there is no mention of the types of underlying infections and the study refers to only nosocomial infections. We have added this and we now cite the primary study as well the review paper. We added some other primary studies, which refer to general issues with patients needing critical care. It does seem to us that we could point to patients with HIV cancer or cardiovascular underlying conditions or probably SARS and complications from fungal colonization. However, It seems more useful to follow the clinical studies, which mostly are concerned with the generality of the susceptibility of critically ill patients. We rewrote the text as requested as follows:

*" However, fungal infections are of great importance due to complications in other diseases.⁵ For example recent studies (in Canadian ICUs) showed upto 50% of critically ill patients were colonised with *Candida* sp.^{6,7} and mortality rates as high as 60% in nosocomial infections have been reported.^{6,8,9}"*

2. Page 3. Line 34 onward: The authors move to a personal tense (e.g. "We") – the editor can recommend if this should be moved to an impersonal tense.

Answer: Our view is that the use of the first person here aides the flow of the text. We will switch this to third person if required but this section appears to be more "readable" in the first person.

3. Page 5, Line 49: The authors state that the coil-globule transition occurs at the LCST. At this point the use of LCST and T_c-g seems to vary in the manuscript. E.g. "...increase in T_c-g; from a peak of 18.9 °C for the as prepared

pyrrole-functional material” compared to “LCST measurements were carried out on a Nano DSC by TA Instruments” (SI, 3.). LCST used in the methods section in the main paper. If these are used interchangeably, I think a single designation would be clearer.

Answer: There is a subtle difference in that LCST refers to the empirical observation of phase separation and T_{c-g} is the temperature at which the desolvation occurs; usually they are both coincident. However, we think this is a good point and there is no need to use the term LCST in this work. We have removed the definition of LCST and we now use T_{c-g} throughout.

4. Page 7, Section “HB-PNIPAM-AmB is active against fungi” – it would be useful to clarify if the MIC is calculated based on the moles of AmB in the polymer, or the moles of polymer itself. I assumed moles of AmB in the polymer-AmB solution, but then the cytotoxicity experiments, which I assumed were conducted based on polymer concentration, asked for comparison to the MICs (“even at 5 mg mL⁻¹, which is far in excess of the MIC data”). I think this confusion arises because the weight-based MIC appears to be related to polymer concentration, whereas the molar MIC is based on conc of AmB in the polymer. I guess this is the case because the only way I could get the numbers to make sense is convert weight to moles using this conversion factor given in the table in figure 1. I recommend making this conversion clear in the table headings and caption.

Answer:

*1. We added the following to Table 1 " * The weight based MIC is the total mass per unit volume including the polymer and the amphotericin end groups. # The molar MIC is the concentration of Amphotericin (mol dm⁻³) without regard for the attachment to the polymer."*

2. In the sentence identified we added to the text as follows (shown in red);

*“even at 5 mg mL⁻¹, which is far in excess of the **weight based** MIC data”).*

3. In Figure 3 we added:

" All concentrations of HB-PNIPAM-AmB are weight of total polymer per unit volume. "

5. Page 7. “AmB compared with the cell only control (... 100 ± 5.2 %)” It is not very clear to me from the methods how you determined error on these measurements of control if the data are normalised to these healthy cells.

Answer: We did this via propagation of errors and have added the following to the experimental (Rabbit limbal epithelial cell viability; Human Renal epithelial cell viability; Human excised corneas; Cell viability testing of PBMCs).

" The variances were calculated via propagation of errors following calculation of the percentage viability."

6. Page 8, figure 3. The caption is written in a confusing manner. “Cell viability, relative to untreated controls, either in the absence of polymer or in the presence of AmB:” I guess these comments relate to the controls, but it is written in a confusing manner. Then for A,B and D,E the two test samples are separated, but placed together in C. I would suggest revising the figure caption and splitting element C to make this easier to follow for the reader.

Answer:

1. We removed "either in the absence of polymer or in the presence of AmB ", which we agree in hindsight is confusing.

2. We split the data in Figures A,B and D,E because the scales of the abscissas are very different and differ by several orders of magnitude. To represent the data most effectively requires two separate graphs. However, the data in C fit comfortably onto a single abscissa. We are of the view that this is the most effective way to present the data.

7. Page 17. “HB-PNIPAM-AmB both bound to ergosterol and was effective against a range of bacteria” – I assume this is meant to read “fungi”

Answer: we corrected this to fungi. The referee is quite correct, this is typographical error.

8. Page 18. “Synthesis of HB-PNIPAM-AmB”: “HB-PNIPAM with COOH end groups prepared as previously described.³⁶” – I understand that this synthesis is described in the SI under “Synthesis of HB-PNIPAM Precursor.” It would be easier to follow with a reference to the SI and a consistent terminology used for this polymer. This method could also include the LCST of the final product, for consistency.

Answer: We name the precursors through out as HB-PNIPAM-Py or HB-PNIPAM-COOH, as appropriate, in the new documents and we added a reference to the SI. We also improved clarity by moving the details of the synthesis of HB-PNIPAM to SI and we provide a reference to section in the SI. We also noted that we already provided the LCST/ T_{c-g} of HB-PNIPAM-COOH in experimental, $T_{c-g} = 22.2$ °C.

9. Page 19 “Overnight cultures were adjusted to an optical density at 600 nm of 0.1 and incubated with Amphotericin polymer, which was serially diluted 1:2 from 2500-2.44 $\mu\text{g mL}^{-1}$, for 16 hours in a 96 well plat” – it would be useful to state the incubation temperature (I guess 37C), especially given that the LCST seems to be very close to 37C. I would imagine the LCST transition may affect MIC.

Answer: We would propose that the LCST does effect the MIC but more detailed work is required in this area. The incubation temperature has now been added (37 °C).

10. Page 20. “Rabbit limbal epithelial cell viability” and “Human Renal epithelial cell viability” – whether data are compared using one or two way ANOVA should be stated.

Answer: We added this; one way ANOVA was used to compare the differences as the concentration changed.

11. Page 21, Conclusions. “In conclusion, highly-branched poly(N-isopropyl acrylamide) with AmB end groups binds to ergosterol with a desolvation phase transition”. This implies to me that the ergosterol binding to PNIPAM-AmB induces a phase change of the whole material (i.e. the coil-globule transition), when I understand there is really some dehydration of AmB end groups. I suggest this statement is revised.

Answer: This is a very good point and we have modified this section to " In conclusion, highly-branched poly(N-isopropyl acrylamide) with AmB end groups binds to ergosterol and provides a desolvation phase transition of a fraction of polymer segments. "

Reviewer: 2

Comments to the Author(s)

This is a really interesting paper and i enjoyed reading it. I am recommending publication with minor cosmetic revisions.

P2, L3 "less toxic" than what"? Recommend this single sentence be more specific.

Answer: The sentence is :

" A branched polymer with amphotericin end groups is active against fungi but is less toxic than Amphotericin-B." so this indicates the polymer is less toxic than Amphotericin-B.

Figure 1: The use of a,b,c in superscript was difficult to see and could be confusing to non-specialists (who may not understand M_w^a is actually M_w with an annotation. Recommend using *,\$,# instead.

Answer: this is a good point and we have changed the superscripts to be less confusing.

Figure 1: Move in figure text into Figure 1 legend.

Answer: this has been changed as requested

Table 1: Legend above table not below.

Answer: this has been changed as requested

Figure 3: legend. $P < 0.001$ is missing its ***

Answer: this has been added as requested

Figure 4: Check colours of * vs those on figures. Consider rewording the significance sentence as it is a little confusing (suggest removing colours and just use *, **, *** in black and description). See Figure 5 for clear example. I think reader will realise that the * are colour coded to lines in the figures.

Answer: We agree the way this was written was confusing. What we are attempting to do is to simplify the representation of various significant differences. We use black to indicate that all samples at a time point are significantly different to the peroxide control. Doing that makes the figures easier to read and less busy. Then we use colour to indicate where individual concentrations are significantly different to the peroxide control.

We have changed the txt for the figures as follow and hope this is now clear.

Figure 4 Relative metabolic activity (MTT, optical density of sample/optical density of untreated control) of PBMCs in contact with HB-PNIPAM-AmB. PBMCs derived from four anonymous donors. The data are relative to cell-only controls and compared to H_2O_2 positive controls. * indicate significance. The black * indicate that all of the HB-PNIPAM-AmB concentrations were significantly different to the H_2O_2 control. The coloured *, *, * indicate that individual concentrations of HB-PNIPAM-AmB were significantly different to the H_2O_2 . * $p < 0.05$, ** $p < 0.01$, *** $p < 0.001$.

Figure 6 Release of TNF- α following treatment with HB-PNIPAM-AmB Concentrations of cytokines in supernatants (cells seeded at 20,000 cells per well) up to 24 hours after treatment with HB-PNIPAM-AmB or LPS. * and + indicate significance. The black * indicate that all of the HB-PNIPAM-AmB concentrations were significantly different to the LPS control at the stated time point. The coloured *, *, * indicate significant differences between individual concentrations of HB-PNIPAM-AmB with LPS. * $p < 0.05$, ** $p < 0.01$, *** $p < 0.001$. The black + indicate that all of the HB-PNIPAM-AmB were significantly different to the untreated cells at the stated time points. The coloured +, +, + indicate significant differences between individual concentrations of HB-PNIPAM-AmB and LPS. + $p < 0.05$, ++ $p < 0.01$, +++ $p < 0.001$.

Figure 7 Release of IL-1 β following treatment with HB-PNIPAM-AmB Concentrations of IL-1 β in supernatants (cells seeded at 20,000 cells per well) up to 24 hours after treatment with HB-PNIPAM-AmB or LPS. There is no significant increase in the release of IL-1 β following treatment with 0.5 mg mL⁻¹ of HB-PNIPAM-AmB but concentrations of over 5 mg mL⁻¹ stimulated release of increase amounts of this cytokine. * and + indicate significance. The black * indicate that all of the HB-PNIPAM-AmB concentrations were significantly different to the LPS control at the stated time point. The coloured *, *, * indicate significant differences between individual concentrations of HB-PNIPAM-AmB with LPS. * $p < 0.05$, ** $p < 0.01$, *** $p < 0.001$. The black + indicate that all of the HB-PNIPAM-AmB were significantly different to the untreated cells at the stated time points. The coloured +, +, + indicate significant differences between individual concentrations of HB-PNIPAM-AmB and LPS. + $p < 0.05$, ++ $p < 0.01$, +++ $p < 0.001$.